# Updated SO2 emission estimates over China using OMI/Aura observations

**Maria Elissavet Koukouli[1], Nicolas Theys[2], Jieying Ding[3,4], Irene Zyrichidou[1], Bas Mijling[3], Dimitrios Balis[1] and Ronald Johannes van der A[3]**

[1]Laboratory of Atmospheric Physics, Aristotle University of Thessaloniki, Greece.

[2]Royal Belgian Institute for Space Aeronomy (BIRA-IASB), Brussels, Belgium.

[3]Royal Netherlands Meteorological Institute (KNMI), De Bilt, The Netherlands.

[4]Technical University Delft, Delft, The Netherlands

## Abstract

The main aim of this paper is to update existing sulphur dioxide (SO2), emission inventories over China using modern inversion techniques, state-of-the-art chemistry transport modelling (CTM), and satellite observations of SO2. Within the framework of the EU FP7 *Monitoring and Assessment of Regional air quality in China using space Observations*, MarcoPolo project, a new SO2 emission inventory over China was calculated using the CHIMERE v2013b CTM simulations, ten years of OMI/Aura total SO2 columns and the pre-existing Multi-resolution Emission Inventory for China (MEIC v1.2). It is shown that including satellite observations in the calculations increases the current bottom-up MEIC inventory emissions for the entire domain studied [102° to 132°E and 15° to 55°N] from 26.30 Tg/annum to 32.60 Tg/annum, with positive updates which are stronger in winter [~36% increase]. New source areas were identified in the South West [25-35°N and 100-110°E] as well as in the North East [40-50°N and 120-130°E] of the domain studied as high SO$_2$ levels were observed by OMI, resulting in increased emissions in the aposteriori inventory that do not appear in the original MEIC v1.2 dataset. Comparisons with the independent Emissions Database for Global Atmospheric Research, EDGAR v4.3.1, show a satisfying agreement since the EDGAR 2010 bottom-up database provides 33.30 Tg/annum of SO2 emissions. When studying the entire OMI/Aura time period [2005 to 2015 inclusive], it was shown that the SO2 emissions remain nearly constant before year 2010 with a drift of -0.51±0.38 Tg/annum and show a statistically significant decline after year 2010 of -1.64±0.37Tg/Annum for the entire domain. Similar findings were obtained when focusing on the Greater Beijing Area [110° to 120°E and 30° to 40°N] with pre-2010 drifts of -0.17±0.14 and post-2010 drifts of -0.47±0.12Tg/annum. The new SO2 emission inventory is publicly available and forms part of the official EU MarcoPolo emission inventory over China which also includes updated NOx, VOCs and PM emissions.

# 1 Introduction

Due to its undoubtable rapid economic growth, swift urbanization and consequent enlarged energy needs, large parts of China have been suffering from severe and persistent environmental issues including major air pollution episodes (Song, et al., 2017.) Developing and implementing effective air quality control policies is essential in combating such pollution problems and requires timely as well as dependable information on emission levels (Zhang et al., 2012; van der A, et al., 2017.) Understanding and monitoring the local long-term trends of different atmospheric pollutants is paramount in updating, and predicting, pollution emission scenarios (Kan, et al., 2012.) Satellite atmospheric observations have recently become an important information source for the atmospheric state, not only of the academic community, but also by public authorities and international environmental agencies (Streets et al., 2013; Lu and Liao, 2016). Recent reductions of the two major pollutants emitted mainly by industrial sources, nitrogen and sulphur dioxide, have already successfully been observed and quantified from space-born instruments over China (Wang et al., 2010; 2015, Liu et al., 2015; 2017).

Sulphur dioxide, $SO_2$, is released into the atmosphere through both natural and anthropogenic processes. In the former category lie chemical processes, such as the reaction of hydrogen sulfide, which is naturally occurring in crude petroleum and natural gas as well as from the breakdown of organic matter, with the atmospheric oxygen, seasonal biomass burning events, which may be foreseen to some extent, if not modelled, as well as volcanic degassing and unexpected eruptions (see for e.g. Seinfeld and Pandis, 1998). In the latter category fall the combustion of coal and oil fuel which account for more than 75% of global $SO_2$ emissions (Klimont et al., 2013), a figure found to be similar when focusing on the Chinese domain (Smith et al., 2001; 2011). Lu et al., 2011, showed that $SO_2$ emissions over China, calculated from all major anthropogenic sources as well as scheduled biomass burning events by the agricultural sector in order to clear vegetation and rejuvenate croplands, increased from ~24 Tg in year 1996 to ~31 Tg for year 2010, including fluctuations due to the onset of environmental protection measures as well as the international economic crisis. The balance between encouraging China's economic development and dealing with its environmental side-effects often causes irregular changes in the $SO_2$ emitted amounts, further dependent on the Province observed.

Satellite $SO_2$ observations have proven to be a reliable way to monitor emissions from space and are increasingly used in order to update bottom-up emission inventories (Streets et al., 2013). Numerous works have already amply demonstrated the ability of satellite sensors to observe regional anthropogenic emission sources such as studying the $SO_2$ load over China using OMI/Aura observations. Krotkov et al. 2016, have shown how using long-term atmospheric data records from the same instrument [OMI/Aura] can provide consistent spatiotemporal coverage enabling the analysis of both anthropogenic and natural emissions. For the North China Plain, of direct interest to this work, it was show that while exhibiting the World's most severe $SO_2$ pollution, since 2011 a decreasing trend with a 50% reduction in emissions has been verified from space. It is of course not only the varying economy and enforcing legislation that affects air quality; Witte et al., 2009, calculated a 13% reduction in sulphur dioxide emissions due to strict pollutant control for the August-September 2008 Olympic and the Paralympic Games held in Beijing observed from space. Li et al., 2010, further demonstrated that the OMI/Aura observations are capable of verifying the effectiveness of China's

SO$_2$ emission control measures on power plants while the imbalance in coal consumption between the different provinces in China was also shown by Jiang et al., 2012. This inter-province diversion was further examined in van der A, et al., 2017, who showed how provinces enforcing desulphurization devices on their power plants have a decreasing SO2 trend whereas emerging provinces, which built new power plants to accommodate the rapid urbanization of the Chinese population, contribute with high emissions to the country's estimates.

Quite recently a new technique uses OMI/Aura observations as means to detect large point SO2 emission sources from diverse origins presented by Fioletov et al., 2013; 2016. Satellite observations were used not only to identify but also to group SO2 emissions into emissions by volcanoes, power plants, smelters, oil and gas industry. The technique has been evolved [Fioletov et al., 2017] into directly assessing traditional statistically-obtained emission levels using OMI as well as OMPS/NPP SO2 columns, with excellent validation results.

Following the aforementioned findings, in this work we aim to present a new spatially-resolved SO2 emission inventory on a monthly time scale for years 2005 to 2015 based on satellite observations and modern chemical transport modelling simulations. The technique used here has recently been applied in both Europe [Zyrichidou et al., 2015] as well as China [Gu et al., 2014] for NOx emissions based on both GOME/ERS-2 and OMI/Aura observations. We aim at showing how it can be applied also to SO2 emissions, and how the new, top-down emissions, compare against traditional bottom-up emission inventories.

# 2 Data Description

The mathematical analysis used in this work in order to extract an updated SO$_2$ emission inventory is fully described in Section 3. The main gist is that three inputs pieces of information are required; an original, also known as *apriori*, emission inventory, the satellite observations of the SO$_2$ load and SO$_2$ profiles provided by an air quality chemistry transport model. The quality of these three pieces of information ensures the accuracy of the updated, *aposteriori*, SO$_2$ emissions estimates. Since the mathematical formulism requires also quantifiable error estimates on these three input parameters, using the new OMI/Aura BIRA SO2 dataset [Theys et al., 2015; 2017] ensures that the satellite observations used here are fully characterized in this manner. In Sections 2.1 to 2.3 the three input datasets are presented and discussed appropriately.

## 2.1 The MEIC emission inventory

The Multi-resolution Emission Inventory for China (MEIC v1.2) model has been developed for years 2008, 2010 and 2012, by the School of Environment, Tsinghua University, Beijing, China and is downloadable from http://www.meicmodel.org/. SO$_2$ emissions, in Mg/month, are calculated on a monthly basis for four sectors: power, industry, residential, and transportation, in a spatial resolution of 0.25x0.25 degrees. The domain applicable spans from 102°E to 132°E and from 15°N to 55°N. For the

requirements of the methodology applied here the error on these emissions has been assumed to rise to 50% of the actual reported value since the MEIC inventory does not include such an error estimate, nor were we able to procure such a value from literature.

An example of the $SO_2$ MEIC v1.2 emissions in Mg/month for March 2010 is shown in Figure 1.The relative strength of the four sectors is shown as well, with industry on the top left panel, the power sector on the top right, the residential emissions in the bottom left and transportation in the bottom right. Different colour scales in the panels were used for the different emission strengths. In Zhang et al., 2015, the 2010 MEIC v1.2 emissions have been used as spin-up information in order to perform sensitivity simulations with different $SO_2$ emission reduction scenarios. It was shown that reducing $SO_2$ emissions from one region has a small effect on $SO_2$ concentrations over the other regions. The national mean $SO_2$ concentration however is most sensitive to $SO_2$ emissions from Northern China, in this work called the Greater Beijing Area. This strengthens the importance of providing accurate and updated emission levels over that region in China even though it is considered to be the best represented within existing inventories since the large population and industry density renders the evaluation of emission levels easier than at remote, less populated, regions.

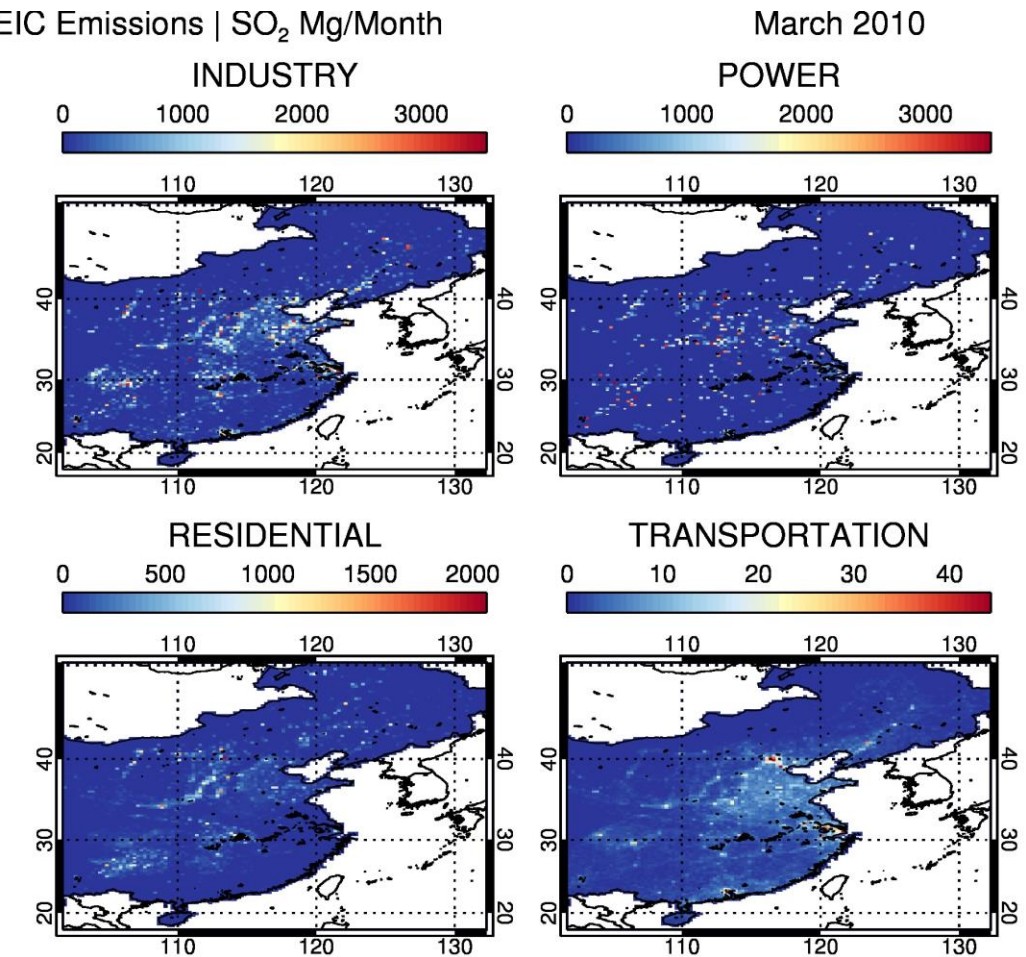

Figure 1. The SO$_2$ MEIC v1.2 emissions in Mg/month for March 2010. The relative strength of the four sectors is shown here; industry, top left; power, top right; residential, bottom left and transportation, bottom right. Note the different colour bars used.

## 2.2 The OMI/Aura SO2 observations

The Ozone Monitoring Instrument (OMI) is a nadir-viewing instrument on board the NASA Aura satellite flying in a Sun-synchronous polar orbit with an equator crossing time of around 13:30 local time in the ascending node launched in July 2004. The OMI imaging spectrograph measures backscattered sunlight in the ultraviolet-visible range from 270 nm to 500 nm with a spectral resolution of about 0.5 nm [Levelt et al., 2006]. The OMI spatial swath is around 2600 km wide achieving near-complete global coverage in approximately one day. The OMI ground pixel size varies from 13 × 24 km$^2$ at nadir to 28 × 150 km$^2$ at the edges of the swath. Since June 2007, the radiance data of OMI for some particular viewing directions have been corrupt, a feature known as the *OMI row anomaly*

(http://www.knmi.nl/omi/research/product/rowanomaly-background.php). Hence, the suggested
OMI observations are excluded de facto from the analysis.

In this work, we employ the retrieved SO2 Vertical Column Densities (VCDs) using the Royal Belgian
Institute for Space Aeronomy, BIRA, algorithm [Theys et al., 2015] which are calculated using the
Differential Optical Absorption Spectroscopy (DOAS) technique [Platt and Stutz, 2008] to the
measured spectra in the 312–326 nm wavelength range. This step is followed by data filtering for the
row anomaly issue and a background correction to account for possible biases on the retrieved slant
columns. The obtained quantity is converted into a SO2 VCD using an air mass factor, AMF, which
accounts for changes in measurement sensitivity due to observation geometry, ozone column, clouds,
and surface reflectivity. The anthropogenic SO2 profile required in the AMF calculation has been
extracted from the Intermediate Model of the Global and Annual Evolution of Species, IMAGESv2,
global tropospheric chemistry transport model [Stavrakou et al., 2013, and references therein] on a
daily basis and for the overpass time of OMI. All details on the BIRA OMI SO2 algorithm can be found
in Theys et al. [2015] updated recently in Theys et al., 2017 in preparation for the TROPOMI instrument.
The dataset has already been employed in different studies; in van der A et al. [2017] in order to
estimate the effectiveness of current air quality policies for SO2 and NOx emissions in China; in
Koukouli et al., 2016, in order to quantify the anthropogenic SO2 load over China using different
satellite instruments and algorithms; in Schmidt et al., 2015, in order to study the 2014–2015
Bárðarbunga-Veiðivötn fissure eruption in Iceland, among others.

The domain considered extends from 102° to 132°E and from 18° to 50°N and covers Eastern China. Daily
observations were filtered for high Solar Zenith Angle, SZA, of > 70°, cloud fraction of > 0.2 as well as
row anomaly flagging as per Theys et al. [2017]. The filtered data were then averaged onto a 0.25°x0.25°
monthly grid using a 0.75° smoothing average box. For further details on this pre-processing refer to
Koukouli et al. [2016].

Within the OMI BIRA $SO_2$ product, error contributions resulting from each step of the retrieval to the
final vertical column error are provided separately, including their random and systematic parts [Theys
et al., 2017]. This allows the estimation of the total error on the column averages, an important feature
in this analysis where the instantaneous OMI observations are gridded and then averaged on a
monthly mean basis. The formulation of the error on the vertical $SO_2$ column is derived by basic error
propagation, shown in Eq. (1).

$$\sigma_{N_V}^2 = \left(\frac{\sigma_{N_S}}{M}\right)^2 + \left(\frac{\sigma_{N_S^{\text{back}}}}{M}\right)^2 + \left(\frac{\left(N_S - N_S^{\text{back}}\right)\sigma_M}{M^2}\right)^2 \tag{1}$$

where $\sigma_{N_S}$, $\sigma_M$ and $\sigma_{N_S^{back}}$ are the errors on the slant column, $N_S$, the air mass factor, M, and $N_s^{back}$ the
reference correction, respectively. When averaging the observations, the systematic and random
components of each given error source need to be discriminated and so Eq. (1) evolves into Eq. (2)

$$\sigma_{N_V}^2 = \frac{1}{M^2}\left(\sigma_{N_S\_syst}^2 + \frac{\sigma_{N_S\_rand}^2}{N} + \frac{\Delta N_S^2}{M^2}\sigma_{M\_syst}^2 + \frac{\Delta N_S^2}{M^2}\frac{\sigma_{M\_rand}^2}{N}\right) \qquad (2)$$

where $N$ is the number of ground pixels considered in the average and $\sigma_{N_S\_syst}$ is the systematic uncertainty on the slant column density, SCD, which also includes the systematic uncertainty associated to the background correction. The Vertical Column Density, VCD, is denoted by Nv; the SCD by Ns; the SCD minus the SCD_correction by ΔNs; the AMF by M; the VCD precision by $\sigma_{NV}$; the SCD precision by $\sigma_{NS\_rand}$; the AMF precision by $\sigma_{M\_rand}$ and the AMF trueness by $\sigma_{M\_syst}$. The error analysis is accompanied by the total column averaging kernel (AK) calculated as the weighting function divided by the air mass factor, M [Eskes and Boersma, 2003]. The weighting function characterizes the sensitivity of the extracted atmospheric column to changes in the true profile and its importance in the analysis of satellite observations, alongside their correct comparison to other datasets, has long been established [see for e.g. Rodgers 2000, Ceccherini and Ridofli, 2010, Zhang et al., 2010, etc. ] In Section 2.3 the importance of the AKs in co-analyzing satellite observations and modelling results in this work is discussed extensively.

An example of the OMI SO2 product used in this work is shown in Figure 2, for the month of March 2010. The retrieved SO2 VCD in Dobson Units (D.U.) is shown in the upper panel with the systematic component to the error in the bottom left and the random component in the bottom right.

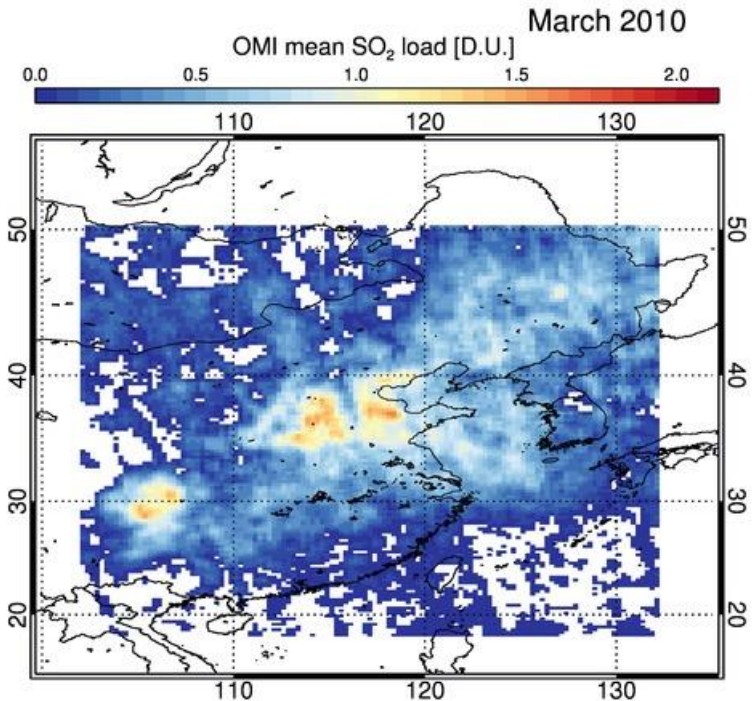

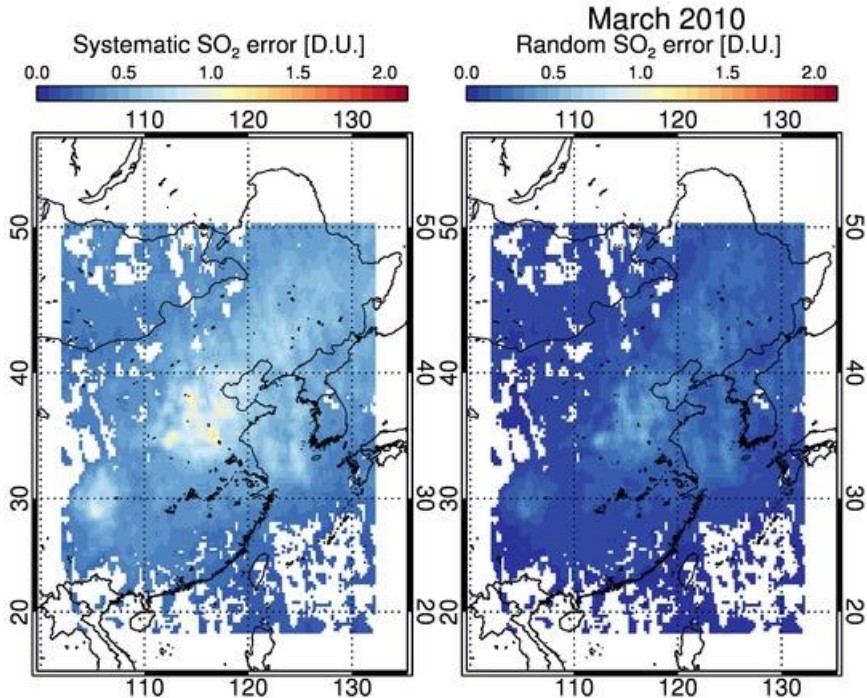

Figure 2. Upper panel: the monthly mean OMI/BIRA SO2 columns in D.U. for March 2010. Lower panel: the associated systematic error [left] and random error [right] in D.U. calculated using Eq. (2).

In the original work of Martin et al., 2006, which was based on GOME/ERS-2 observations and GEOS-CHEM model data on a resolution of 2° by 2.5°, the authors conclude that the major limitations in their work were the coarse horizontal resolution of GOME – which is not the case here for OMI– and the lack of direct validation of the GOME tropospheric NO2 product – again, not this case here as the OMI BIRA SO2 measurements have already been verified against other satellite observations [Bauduin et al., 2016; Koukouli et al., 2016] as well as long term ground-based measurements in polluted locations [Theys et al., 2015, Wang et al., 2017]. However, we would be amiss not to mention the issue of the possible horizontal transport of SO2 during its lifetime in the lower troposphere which would alter the linear relationship inherent in Eq. (4). Hains et al., 2008, calculated the SO2 lifetime on a global scale to be 19 ± 7 h, whereas Lee et al., 2011, have updated this estimate, at northern US mid-latitudes where anthropogenic emissions dominate, to 16–40h with a maximum in winter and a minimum in summer. Using OMI/Aura observations over the highest emitting power plant locations in the US, Fioletov et al., 2015, have provided a shorter lifetime estimates of between 4 and 12 h. Even though it is hence not inconceivable that with moderate wind speeds SO2 may have traversed a grid point on our 0.25°x0.25° grid, on the monthly mean scale that this work is based on it is impossible to evaluate the magnitude to this possible smearing effect.

## 2.3  The CHIMERE model output

A multi-scale model for air quality forecasting and simulation, CHIMERE, http://www.lmd.polytechnique.fr/chimere/, is providing $SO_2$ profiles over the Chinese domain between 102°E - 132°E and 18°N - 50°N for the mean overpass hour of OMI/Aura over the domain. The model version is CHIMERE v2013b [Menut et al. 2013] at a spatial resolution of 0.25°x0.25° and on eight vertical levels in ppb, i.e. seven vertical layers, spanning from the surface up to 500hPa, for year 2010. The meteorological input was provided by ECMWF, http://www.ecmwf.int/, operational data. The anthropogenic emission inventory in this CHIMERE run was a mix of the MEIC v1.2 inventory for mainland China and the Intex-B emission inventory, http://mic.greenresource.cn/intex-b2006 for areas outside China. The biogenic emissions are provided by the MEGAN database, http://lar.wsu.edu/megan/. For the background of the particular CHIMERE set-up refer to Mijling and van der A, (2012), whereas more specific details on the CHIMERE v2013b run used here may be found in Ding et al., (2015).

The uncertainty of the CHIMERE SO2 columns is assumed to rise to 25%. Estimating mathematically modelling errors is quite challenging due to the large number of modelling processes and input parameters that have no defined error, such as for e.g. the boundary and initial conditions, the species emissions, rate constant uncertainties, even unresolved aspects of atmospheric physics and chemistry [Deguillaume et al., 2008; Boersma et al., 2016]. Typically such uncertainties are deduced from comparisons to other CTMs [Pirovano et al., 2012] and/or to independent observational datasets [Lee et al., 2009]. Even so, due to the innumerous differences in mathematically expressing atmospheric processes in the former case and between model simulations and observations in the latter case, calculating a definite value remains elusive. In Figure 3, upper, the March 2010, CHIMERE integrated SO2 column is shown as example for the domain in question.

Before proceeding to the CHIMERE profiles convolution to the OMI AKs and subsequent vertical integration, we investigated whether the differences in orography heights assumed by the CHIMERE and OMI datasets in the respective algorithms may introduce artifacts in the final CHIMERE VDCs. Zhou et al., 2009, have shown that, for the case of NO2 profiles retrieved from OMI measurements over the Po Valley and the Alps, the difference in orography between satellite pixel and CTM grid may lead to either over- or under-estimation of the NO2 VCDs by between 10 and 25%. Theys et al., 2017, in order to utilize more realistic *apriori* SO2 profiles, employed CTM model profiles at 1°x1° resolution and used the hypsometric equation (Eq. (3)) to scale them down to the future TROPOMI/S5P 7 km × 3.5 km spatial resolution. In this equation, a new effective pressure, $P_{eff}$, which differs from the model surface pressure $P_{ERA}$, is calculated under the assumption that the surface temperature, $T_{ERA}$, varies linearly with height with a lapse rate of $\Gamma$ = -6.5Kkm$^{-1}$, gas constant of R=287 Jkg$^{-1}$K$^{-1}$ and gravitational acceleration of g = 9.8ms$^{-2}$. This variation depends on the difference between the orography height of CHIMERE, $h_{CHIM}$, and the OMI-reported height per observation, $h_{eff}$. The surface pressure and temperature have been extracted from the ERA-interim dataset, https://www.ecmwf.int/en/research/climate-reanalysis/era-interim, on a daily temporal and 0.75°x0.75° spatial resolution [Dee et al., 2011].

In the case of $SO_2$ anthropogenic emissions, this whole issue may be significant in locations where the surface height alters significantly within our 0.25°x0.25° grid whereupon the OMI pixel may have viewed an entirely different atmospheric state, by more than ~1km in the vertical. In this work and for the entire ten years of OMI observations, only 3% of the entire domain of 15609 grid points show an

over-estimation of $h_{CHIM}$ heights above 500m and less than 0.5% of the grid points show an over-estimation of $h_{eff}$ heights.

$$P_{eff} = P_{ERA}\left(\frac{T_{ERA}}{T_{ERA} + \Gamma(h_{CHIM} - h_{eff})}\right)^{-g}/_{R\Gamma} \qquad (3)$$

Even so, and for completeness sake, the CHIMERE profiles were re-scaled accordingly to the new pressure levels, calculated from $P_{eff}$ and the CHIMERE pressure parameters as applied in Equations 2 and 6 of Zhou et al., 2009. Grid points with associated CHIMERE heights of greater than 1500m, which represent 7.5% of the domain, almost exclusively in the western-most part [west of 110°E] where the Tibetan plateau rises, are excluded from this re-scaling due to interpolation issues. Those pixels are in any case excluded in the analysis for the new emission database further on due to their non-existent SO2 contributions. Overall, the non-seasonally dependent differences found in the CHIMERE columns before and after scaling were of the order to ~10-12%, on the low side of the Zhou et al., 2009, estimates for NOx who were however faced with far greater topological variabilities in the locations of their study. As a consequence, we consider the convolution of modelling profiles to the satellite AK a far more important factor in the solidity of the proposed methodology that anything else.

An extremely small fraction of our domain showed significant variation of above 0.5 D.U. in absolute differences, of less than ~0.05% of the pixels for the entire domain irrespective of month, due to numerical uncertainties introduced by the re-shaping, re-scaling and altering between the different altitude domains of the CHIMERE and OMI profiles. Hence, for the main aim of this paper which is to update the $SO_2$ emission spatial inventory over Eastern China and not to provide absolute SO2 emitted quantities, we deem this difference well within the final emission inventory error budget discussed below in Section 4.1.

We then proceed in convolving the re-scaled CHIMERE profiles with the OMI column averaging kernel as discussed in Eskes and Boersma, 2003 and Boersma et al., 2008a. The CHIMERE model profiles were already in a 0.25°x0.25° monthly grid whereas the OMI observations are daily measurements on a variable pixel size, between 13 × 24 km² at nadir to 28 × 150 km² at the edges of the swath. Hence, the CHIMERE profile for each grid was convolved with each of the corresponding OMI AKs that fall within the same 0.25°x0.25° grid and then averaged [see Figure 3, bottom]. On average, the convolution of the CHIMERE re-shaped profiles with the OMI AKs introduced a seasonally dependent decrease in the SO2 modelled levels, between ~0-5% [for the summer months] and 10-15% [for the autumn-winter months] for the entire domain, as expected.

An example of this entire process is provided in Figure 4 for the grid box 38.0°N, 113.25°E, a location slightly to the West of Greater Beijing Area with a moderate orography height of ~1km. In the left panel the original CHIMERE SO2 profile in 8 levels in ppb is shown in blue, the same profile but in Dobson units per layer is given in red whereas the profile in Dobson units but on the OMI AK levels is given in black since the OMI algorithm performs calculations on a 58 level pressure grid. The y-axis ranges up to ~5 km which is approximately the vertical range of the CHIMERE model. In the middle panel the OMI

AK profile is presented. In the right panel the original CHIMERE profile in Dobson units is shown again in black so as to compare easily to the convolved CHIMERE profile, in olive green. Insert in this panel the total $SO_2$ load in D.U. for the two profiles is also given. The re-shaped CHIMERE total SO2 column is 1.50 D.U. whereas after convolution with the OMI AK it decreases to 0.885 D.U. while the actual load is also re-structured in order to approach the atmosphere sense by the satellite instrument. It is hence shown that even though the total column has not changed the vertical distribution of that column does change to reflect the sensitivity of the satellite observations, which peaks higher up in the boundary layer and lower troposphere.

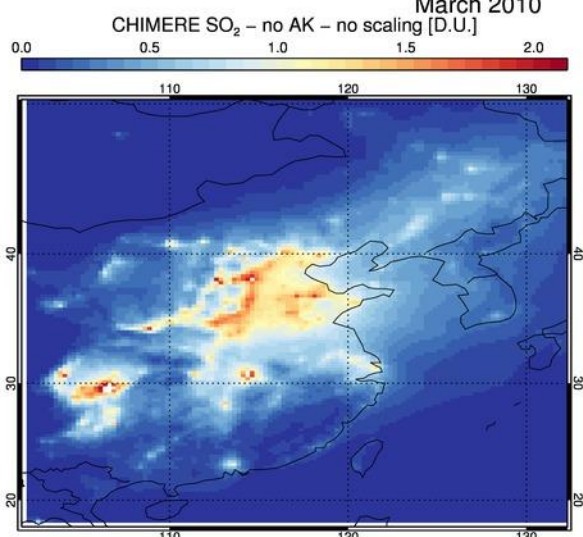

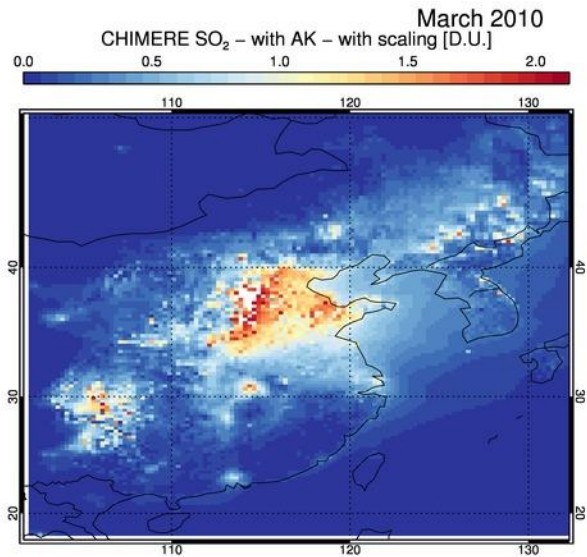

Figure 3. The March 2010 SO$_2$ columns in D.U. as integrated in height from the original CHIMERE model ppb levels: upper, without rescaling to the effective pressure and without convolution with the OMI AKs; lower, with rescaling and with convolution with the OMI AKs.

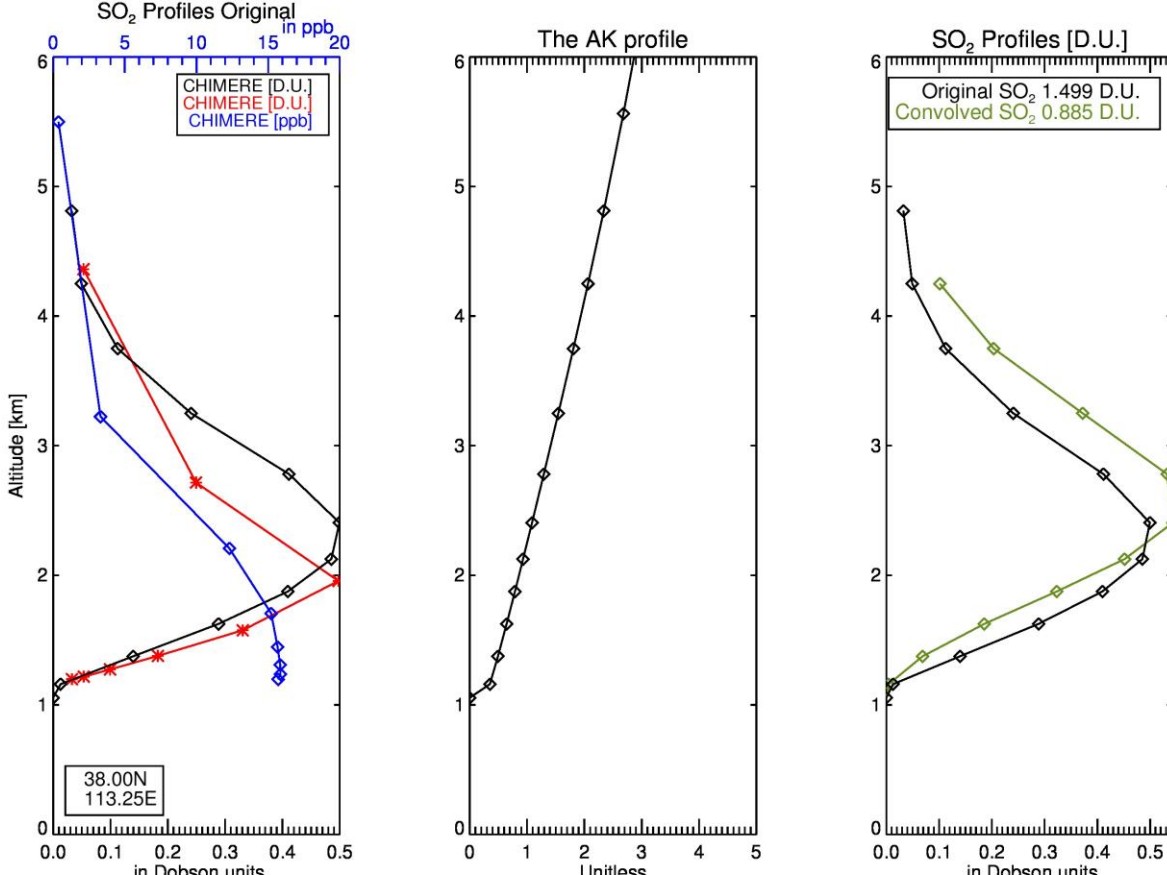

Figure 4. An example of the convolution of the CHIMERE SO2 profile with the OMI Averaging Kernel to produce the convolved CHIMERE total SO2 column for the grid 38.0°N, 113.25°E. Left panel: The original CHIMERE SO2 profile in 8 levels in ppb is shown in blue, the same profile but in Dobson units per layer is given in red whereas the profile in D.U. but on the OMI 58 AK levels is given in black. Middle panel: the OMI AK profile. Right panel: the original CHIMERE profile in D.U. per layer is shown in black, as in the left panel, and the convolved CHIMERE profile is D.U. per layer is shown in olive green. The original CHIMERE total SO2 column is 1.50 D.U. whereas after convolution with the OMI AK it decreases to 0.885 D.U.

# 3 Mathematical nomenclature

## 3.1 Top-down and aposteriori emissions estimates

The inversion methodology applied here is the one first presented in Martin et al., 2003, and further applied in Martin et al., 2006, Boersma et al., 2008b, Lamsal et al., 2010, Lin et al., 2010, Gu et al., 2014, Zyrichidou et al., 2015, among others. The main premise of the methodology resides in the mass balance equation [Leue et al., 2001] and requires three input parameters; the *apriori* emission field, $E_a$ [Sect. 2.1], the satellite-derived $SO_2$ field, $\Omega_t$ [Sect.2.2] and the model $SO_2$ field, $\Omega_a$ [Sect.2.3]. Using those, as per Eq. (4), the *top-down* emission inventory, $E_t$, is calculated. Using standard propagation

error analysis, the error on the *top-down* emission field may be calculated through Eq. (5), where the error on the apriori emissions, $\varepsilon_a$, is required, as well as the error on the model estimates, $\varepsilon_{\Omega a}$ and the satellite retrieval error, $\varepsilon_{\Omega t}$. These error levels have been discussed in the equivalent sections.

$$E_t = E_a * \frac{\Omega_t}{\Omega_a} \qquad (4)$$

$$\varepsilon_t^2 = (\frac{\Omega_t}{\Omega_a} * \varepsilon_a)^2 + (\frac{E_a}{\Omega_a} * \varepsilon_{\Omega t})^2 + (\frac{E_a \Omega_t}{\Omega_a{}^2} * \varepsilon_{\Omega_a})^2 \qquad (5)$$

The calculated *top-down* emission inventory, $E_t$, may be combined with the *apriori* emission inventory, $E_a$, to provide an *aposteriori* emission inventory, $E_p$, following the maximum likelihood theory and a log-normal distribution of errors. In Eq. (6) the calculation of the *aposteriori* emission inventory is given, and its associated relative error in Eq. (7). Hence, in this methodology, the original bottom-up emission inventory is combined with the top-down satellite observations, weighted by their respective errors, and using modeling outputs as background field, in order to constraint, update and provide new emissions estimates. It also follows that since the *apriori* emission field is weighted by the top-down emission field error, and vice versa, the *aposteriori* will depend mostly on the *apriori* should the errors of the top-down be too large, and vice versa. In that way, it is assured that at locations where the satellite observations are too sparse or the information content in the $SO_2$ load too low, the *aposteriori* emission field will revert back to the *apriori.*

$$\ln E_p = \frac{\ln E_a \, (\ln \varepsilon_t)^2 + \ln E_t \, (\ln \varepsilon_\alpha)^2}{(\ln \varepsilon_t)^2 + (\ln \varepsilon_a)^2} \qquad (6)$$

$$(\ln \varepsilon_p)^{-2} = (\ln \varepsilon_t)^{-2} + (\ln \varepsilon_a)^{-2} \qquad (7)$$

We should clarify at this point that the calculations of Eq. (4) to Eq. (6) are performed on domain space, i.e. for completeness sake these equations should have an i.j indicator everywhere designating the lat/lon location of the gridded domain space. The i,j were not included because it was deemed the equations would become too complicated unnecessarily. However, the relative error calculated by Eq. (7), which represents the geometric standard deviation about the expected value as per Martin et al., 2003, is calculated on the final, total top-down error, $\varepsilon_t$, and apriori error, $\varepsilon_a$, which are calculated as the known summation of error terms, $\varepsilon^2 = \varepsilon_{i,j}^2 + \varepsilon_{i,j+1}^2 + \cdots + \varepsilon_{i+1,j}^2 + \varepsilon_{i+1,j+1}^2 + \cdots$.

In the very recent paper by Cooper et al., 2017, an iterative version of the mass balance methodology [Martin et al., 2003] was shown to provide results of similar accuracy as the more computationally demanding adjoint method [used for e.g. in Stavrakou et al., 2013] in estimating satellite-born NOx emissions, which encourages the usage of the mass balance technique when one cannot employ from modelling results that calculate an adjoint matrix as well.

## 3.2 Roadmap of this analysis

The statistical methodology described above will be applied to the entire eleven years of OMI/Aura observations, from 2005 to 2015 inclusive. Since the CHIMERE v2013b simulations were performed using the 2010 MEIC v1.2 inventory, year 2010 will be used as reference year in the following analysis. The first step is to present the 2010 updated emissions over the entire domain and how these compare against the *apriori* emissions; secondly, monthly mean time series of different locations within the domain are shown and the changes of the SO2 emissions over the years is discussed. Finally, comparisons against pre-existing bottom-up emission inventories are presented.

# 4 Results and statistics

## 4.1 Updated emissions over China

In Figure 5 the seasonal variability of the aposteriori emissions calculated with the methodology above are shown in the middle column for spring, summer, autumn and winter [top to bottom.] The equivalent MEIC v1.2 apriori inventory on the same seasonal basis is shown in the left column and the percentage differences of the two in the right column. The main take-away message from this pictorial representation of the inventory is that the new inventory is producing higher emissions for the entire domain for all seasons, which are stronger in winter and have positive biases that span from ~10% to ~35% accordingly [*Table 1*]. Note from the fifth column of the Table the amount of grid points that actually provide information out of an original 8414 grid cells for the domain considered in this work, i.e. the grid cells of the MEIC v1.2 inventory. In the final column of the table, the percentage differences between the two inventories are calculated in two ways: the first value depicts the difference between the first and third columns, i.e. on the sum of emissions for the entire domain. The second value, in square brackets, has been calculated as the mean of the per grid point percentage differences within the domain, hence it contains the geographical deviations of the emission inventories as well. In order to further delve into this geographical variability we present in Figure 6 time series of emissions over four domains of interest; the entire domain studied [18-50°N and 102-132°E], the Greater Beijing region [30-40°N and 110-120°E], the South West region [25-35°N and 100-110°E] and the North East region [40-50°N and 120-130°E]. The two regions in the corners of the area studied were chosen since high $SO_2$ levels were observed by OMI, resulting in increased emissions in the aposteriori inventory, that do not appear in the original MEIC v1.2 dataset.

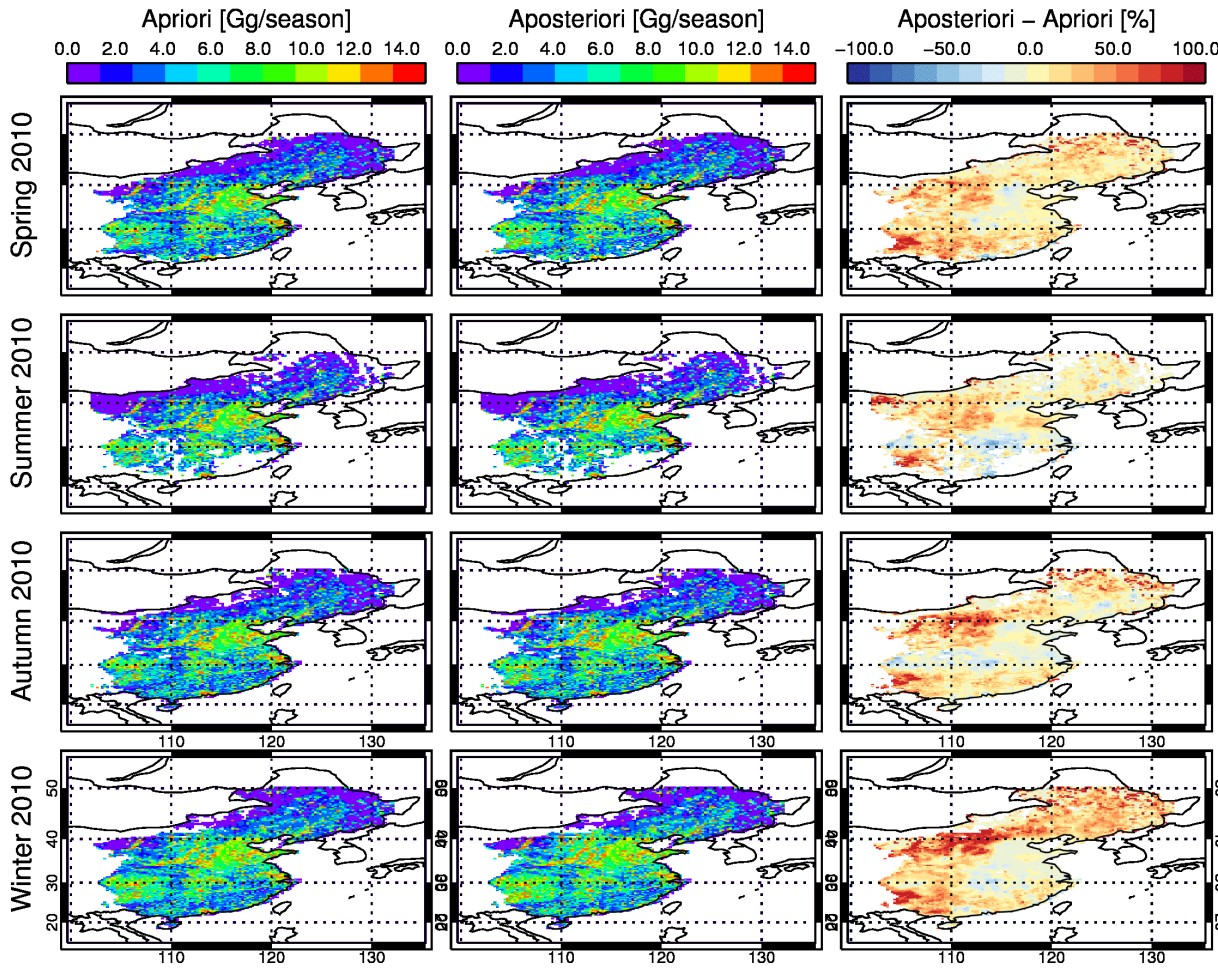

Figure 5. The seasonal variability of the aposteriori emissions calculated in this work [middle column] in Gg/season compared to the apriori MEIC v1.2 emissions [left column] in Gg/season as well as their percentage differences [right column] in %. From top to bottom; spring, summer, autumn and winter of reference year 2010.

Table 1. The average $SO_2$ emission levels over China for the four seasons of year 2010 as presented in Figure 5.

|  | Apriori [Gg/season] | Apriori error [Gg/season] | Aposteriori [Gg/season] | Aposteriori error [Gg/season] | # cells | % difference |
|---|---|---|---|---|---|---|
| Spring | 6.36 | 0.135 | 7.77 | 1.57 | 6975 | 18.0 [24.0] |
| Summer | 5.96 | 0.132 | 6.46 | 1.01 | 5765 | 8.0 [14.0] |
| Autumn | 6.77 | 0.137 | 7.68 | 1.40 | 7126 | 13.0 [20.0] |
| Winter | 7.07 | 0.140 | 9.12 | 2.66 | 7254 | 29.0 [34.0] |

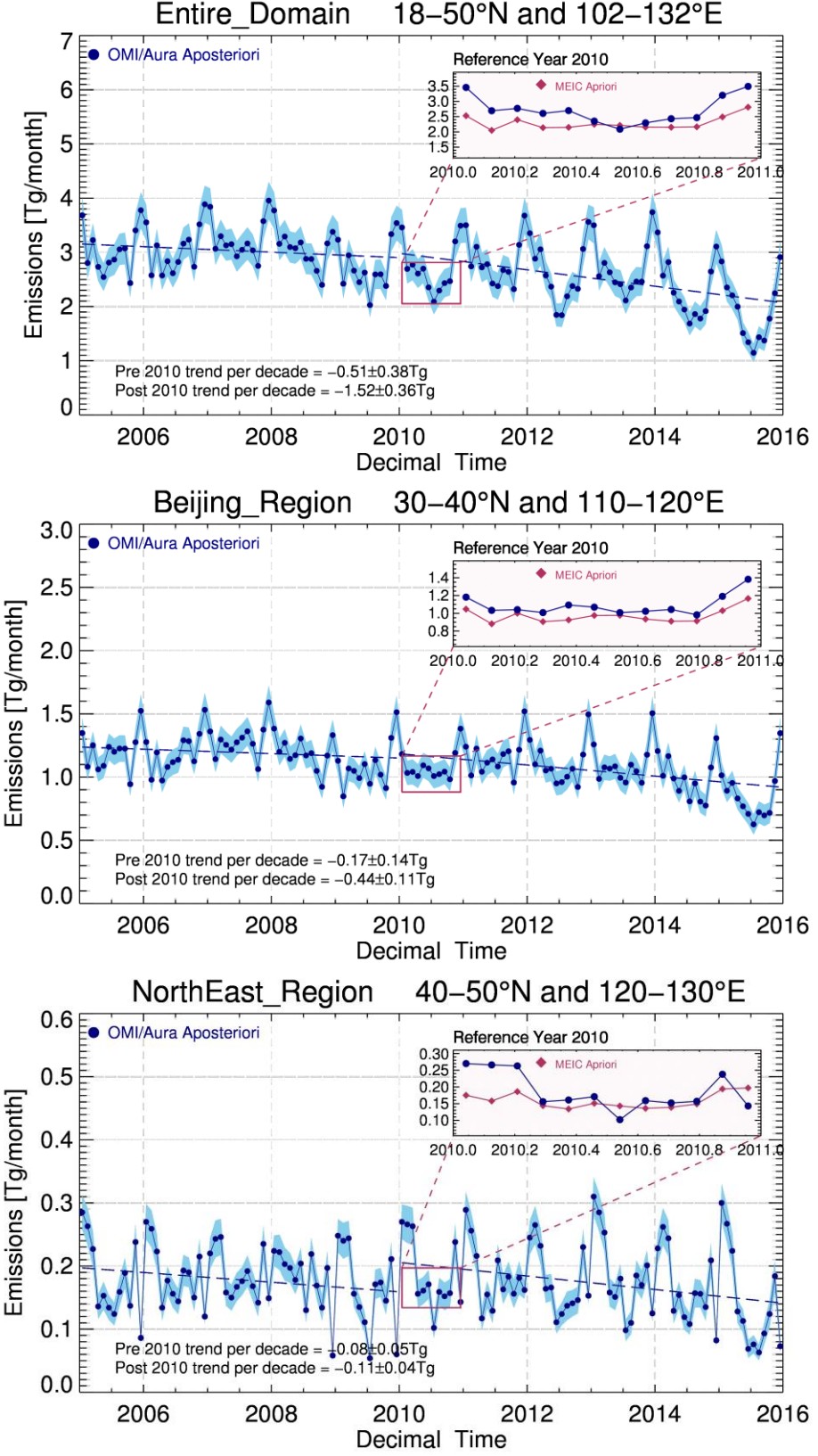

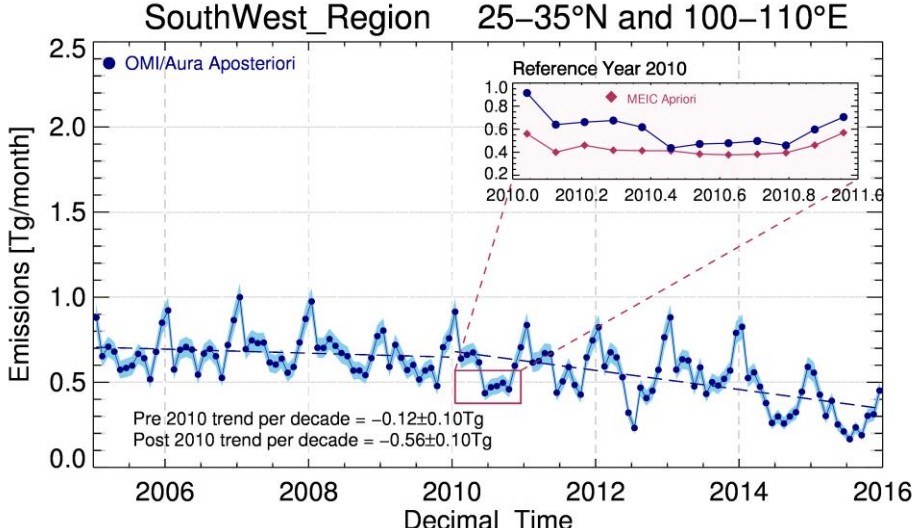

Figure 6. Monthly mean time series for the *aposteriori* emissions in Tg/month calculated in this work [dark blue points] between years 2005 and 2015 inclusive. Insert, the reference year 2010 is shown to include the MEIC v1.2 *apriori* emissions in maroon diamonds. The light blue shaded area depict the calculated apriori error [Eq. (7)]. From top to bottom: the entire domain studied [18-50°N and 102-132°E], the Greater Beijing region [30-40°N and 110-120°E], the North East region [40-50°N and 120-130°E] and the South West region [25-35°N and 100-110°E].

In Figure 6 the monthly mean time series for the *aposteriori* emissions in Tg/month [dark blue lines] are presented for the four domains of interest, so as to enable a more in depth discussion of the new inventory. The light blue shaded area depicts the extracted *aposteriori* error on the emissions and the inset sub-figures depict the reference year 2010 with the *aposteriori* levels shown in blue and the MEIC v1.2 emissions in maroon. The pre- and post-2010 drifts are also calculated since year 2010 is considered a turning point as far as regulating $SO_2$ emissions are concerned [Wang et al., 2015; van der A, et al., 2017, and references therein]. A very similar picture was shown for all domains: a near-stable decrease in emissions within the statistical error of the analysis for the pre-2010 levels and a stronger and statistically significant decrease for the post-2010 levels.

For the entire domain [Figure 6, first panel] aposteriori emissions on all months show an increase for year 2010 compared to the apriori MEIC inventory, apart from the JJA summer ones, with the highest increases for the winter months. The pre-2010 drift is calculated at the limit of the statistically significance, at -0.51±0.38 Tg/month, whereas the post-2010 drift is stronger and significant at -1.52±0.36 Tg/month. For the greater Beijing region [Figure 6, second panel] a small increase in emissions, nearly constant on all months of 2010, is found with the post-2010 drift also negative at the -0.44±0.11 Tg/month level. Two special regions of interest, with low emission levels in general, were revealed by the OMI observations, in the North East and the South West of the domain and are examined in the third and fourth panels respectively. The first three months of year 2010 in the *aposteriori* emission database show quite higher levels that the MEIC v1.2 compilation, whereas the rest of the months show the same level, for the NE whereas in the SE the first six months of the year have an increased SO2 emitting signature.

## 4.2 Comparison with existing emission inventories

Table 2. Details of the existing emission databases used for comparative purposes.

| Database | Years available | Spatial resolution | Temporal resolution | Main reference | Publicly available from: |
|---|---|---|---|---|---|
| REASv2.1 | 2000 to 2008 | 0.25°x0.25° | monthly | Kurokawa et al., 2013 | https://www.nies.go.jp/REAS/ |
| Intex-B | 2006 | 0.5°x0.5° | yearly | Zhang et al., 2009 | https://cgrer.uiowa.edu/projects/emmison-data |
| EDGAR v4.3.1 | 2010 | 0.1°x0.1° | monthly | Crippa et al., 2016 | http://edgar.jrc.ec.europa.eu/ |

Apart from the MEIC v1.2 emission inventory discussed in Section 2.1, which is currently publicly available for years 2008, 2010 and 2012, there exist other emission inventories that are frequently used in chemical transport models as input; the Regional Emission inventory in Asia (REAS) v2.1 [Kurokawa et al., 2013]; the 2006 Asia Emissions for Intex-B [Zhang et al., 2009] and the Emissions Database for Global Atmospheric Research, EDGAR v4.3.1 [Crippa et al., 2016]. Comparing with similar published works is not as straightforward as one would assume since in this work a sub-domain of what is termed *China* in other publications is used. For e.g. when calculating the total annual $SO_2$ emissions reported by the REASv2.1 database for year 2000, those are found to be 25.62Tg per annum when allowing the entire domain provided in the database but only rise to 15.86Tg per annum when restricting in the domain we are studying. As a result, large differences and erroneous comparisons may be presented if one simply compares emissions estimates as reported in published works. For completion purposes we refer the reader to Table 3 of Lu et al., 2010 and Table 8 of Kurokawa et al., 2013, for similar comparative studies, however great care is needed when quoting absolute $SO_2$ emission levels.

In Table 2 the details of the three databases are given. Since we are interested in evaluating the SO2 emission as spatial patterns and not point source levels, we focused on these three databases which provide their databases in actual spatiotemporal resolutions. As a first inspection, in Table 3, the annual SO2 emissions for the domain 102°E - 132°E and 15°N - 50°N in Tg per annum are presented. We should point out that, due to the fact that our methodology is based on the MEIC v1.2 emission inventory, within the domain stated there are large areas with no emissions, mostly over sea and the Korean peninsula. In the following comparisons, only the common pixels between all inventories are used for the calculations naturally.

Several issues arise; firstly, for the common years between this work and the REAS v2.1, i.e. years 2005 to 2008 inclusive, the differences span between ~30 and ~60% with REAS v2.1 underestimating the emission levels in the domain studied. For the one common year between REAS v2.1 and MEIC v1.2, namely 2008, this underestimation still holds but is smaller, of the order of ~10%. Similarly, for the one common year between REAS v2.1 and Intex-B, namely 2006, REAS v2.1 underestimates by ~30%. All these point to an underestimation of SO2 levels in the domain considered by the REAS v2.1 database.

Comparing the 2006 Intex-B emissions to the ones calculated in this work, we find a difference of the order of ~10% whereas comparing to the 2010 EDGAR v4.3.1 emissions the difference is almost

insignificant, at ~3.5%.  Since the EGDAR v4.3.1 emissions are provided on a monthly basis, in contrast to the Intex-B ones, we can evaluate our spatial patterns as well. After regridding the EDGAR v4.3.1 emissions onto a 0.25°x0.25° spatial resolution on a monthly basis, the seasonal variability of the inventory is compared to the one presented in this work in Figure 7.

Table 3. Annual SO2 emissions over the domain 102°E - 132°E and 15°N - 50°N in Tg per annum; first column the year; second column this work; third column the REASv2.1; fourth column, EDGAR v4.3.1 and fifth column, the Intex-B database.

| Year | This work | REASv2.1 | MEIC v1.2 | EDGAR v4.3.1 | Intex-B |
|------|-----------|----------|-----------|--------------|---------|
| Tg/annum for the 102°E - 132°E and 15°N - 50°N domain | | | | | |
| 2000 | | 15.86 | | | |
| 2001 | | 15.94 | | | |
| 2002 | | 17.53 | | | |
| 2003 | | 19.70 | | | |
| 2004 | | 21.77 | | | |
| 2005 | 35.27±1.75 | 24.68 | | | |
| 2006 | 35.33±1.76 | 24.45 | | | 32.08 |
| 2007 | 37.58±1.76 | 24.40 | | | |
| 2008 | 35.75±1.76 | 26.96 | 29.80 | | |
| 2009 | 31.74±1.75 | | | | |
| 2010 | 32.14±1.74 | | 26.26 | 33.34 | |
| 2011 | 33.50±1.75 | | | | |
| 2012 | 31.30±1.75 | | 26.48 | | |
| 2013 | 32.05±1.74 | | | | |
| 2014 | 28.32±1.72 | | | | |
| 2015 | 23.34±1.71 | | | | |

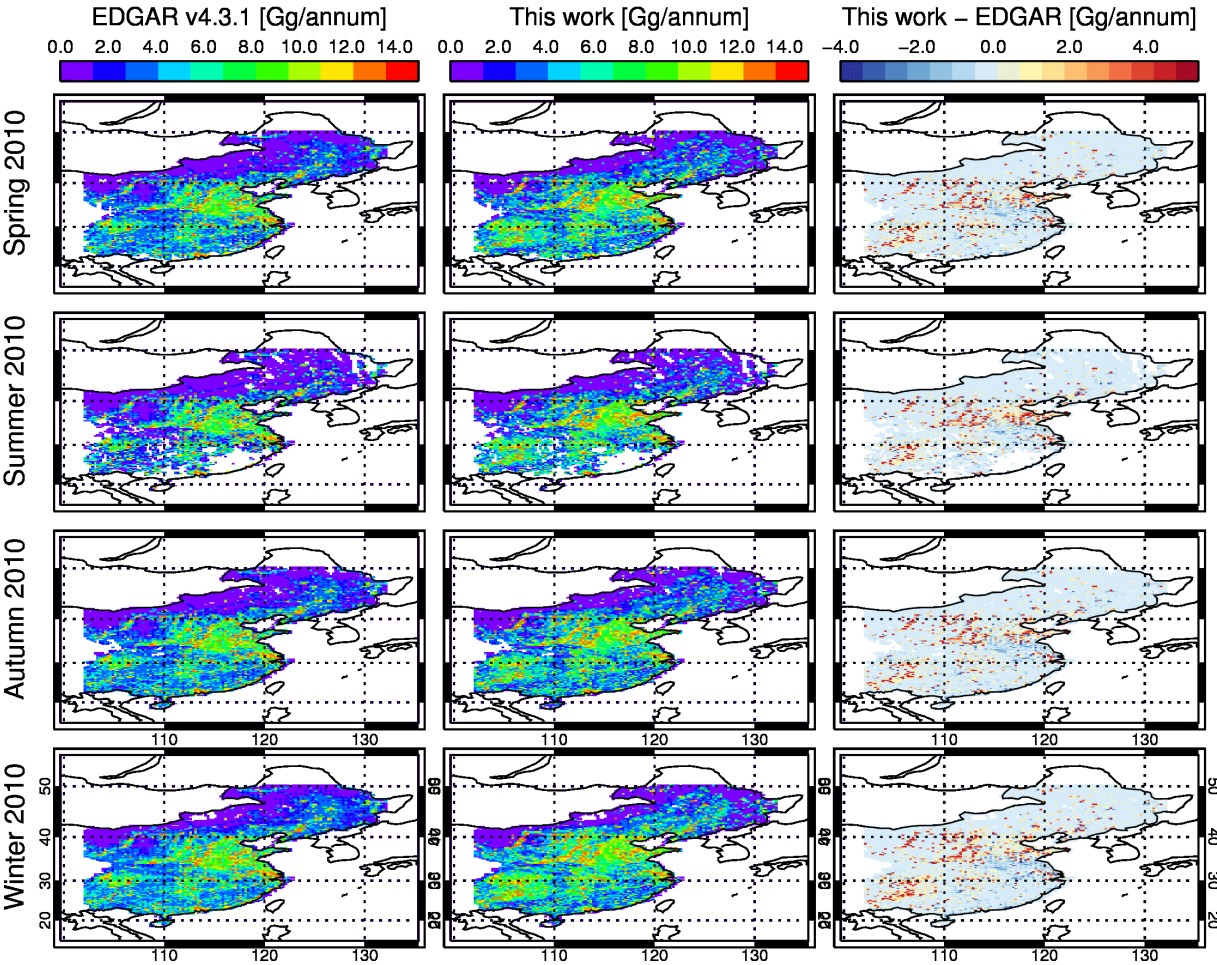

Figure 7. The seasonal variability of the aposteriori emissions calculated in this work [middle column] in Gg/season compared to the EDGAR v4.3.1 emissions [left column] in Gg/season as well as their absolute differences [right column]. From top to bottom; spring, summer, autumn and winter of the reference year 2010.

# 5  Summary

In this work, an updated SO2 emission inventory based on OMI/Aura observations and the CHIMERE v2013b simulations has been presented for years 2005 to 2015 inclusive, as part of the EU FP7 MarcoPolo project which provides updated emissions over China based on satellite observations of key air quality species. For the domain between 102°E - 132°E and 15°N - 50°N it was shown that the annual SO2 emissions calculated remain stable at 36.0±1.0 Tg/annum between years 2005 and 2008, decreasing to 32±0.8Tg/annum between 2008 and 20103, leading to a low of ~23.0 Tg/annum for year 2015, with highs during the winter months and lows during the spring and summer time. Trend analysis performed on the monthly mean spatial averages show that pre-2010, the monthly SO2 emissions were ~3.0±1.0 Tg/month whereas the statistically significant decrease in the post-2010 era rises to -1.52±0.36 Tg. The higher differences to the original apriori MEIC v1.2 2010 inventory were found for the winter

months, especially February, with seasonal differences of the order of ~40% and the smallest for the summer months at ~10%. Comparisons with completely independent emission inventories show a good agreement to the 2010 EDGAR v4.3.1 emissions at the 3.5% level, whereas moderate agreement was found against the 2006 Intex-B database at the ~10% level.

The subsequent logical step in this work is to employ the new emission inventory as input information for a chemistry transport model so as to assess the effect of the updated $SO_2$ emissions on the output simulations, as well as validation against independent sources of information on the point $SO_2$ sources around China, a work under development.

# Data availability

Input datasets:

OMI/Aura $SO_2$ BIRA algorithm, main reference: Theys, N., De Smedt, I., van Gent, J., et al., (2015), Sulphur dioxide vertical column DOAS retrievals from the Ozone Monitoring Instrument: Global observations and comparison to ground-based and satellite data, J. Geophys. Res. Atmos., 120(6), 2470–2491, doi:10.1002/2014JD022657.

CHIMERE v2013b simulations, main reference: Ding, J., van der A, R. J., Mijling, B., Levelt, P. F., and Hao, N.: NOx emission estimates during the 2014 Youth Olympic Games in Nanjing, Atmos. Chem. Phys., 15, 9399-9412, doi:10.5194/acp-15-9399-2015, 2015.

Output datasets:

EU FP7 MarcoPolo $SO_2$ emission inventory is publicly available from [http://www.marcopolo-panda.eu/products/toolbox/emission-data/](http://www.marcopolo-panda.eu/products/toolbox/emission-data/) and the main reference is this work.

Auxiliary datasets:

The MEIC v1.2 database is publicly available from http://www.meicmodel.org/ and the main reference is n/a.

The Intex-B database is publicly available from [https://cgrer.uiowa.edu/projects/emmison-data](https://cgrer.uiowa.edu/projects/emmison-data) and the main reference is Zhang, Q., D.G. Streets, G.R. Carmichael, et al., (2009), Asian emissions in 2006 for the NASA INTEX-B mission, Atmos. Chem. Phys., 9, 5131-5153, doi:10.5194/acp-9-5131-2009.

The EDGAR v4.3.1 database is publicly available from [http://edgar.jrc.ec.europa.eu/](http://edgar.jrc.ec.europa.eu/) and the main reference is Crippa, M., Janssens-Maenhout, G., Dentener, F., Guizzardi, D., Sindelarova, K., Muntean, M., Van Dingenen, R., and Granier, C.: Forty years of improvements in European air quality: regional policy-industry interactions with global impacts, Atmos. Chem. Phys., 16, 3825-3841, doi:10.5194/acp-16-3825-2016, 2016.

The REAS v2.1 database is publicly available from https://www.nies.go.jp/REAS/ and the main reference is Kurokawa, J., Ohara, T., Morikawa, T., Hanayama, S., Janssens-Maenhout, G., Fukui, T., Kawashima, K., and Akimoto, H.: Emissions of air pollutants and greenhouse gases over Asian regions during 2000–2008: Regional Emission inventory in ASia (REAS) version 2, Atmos. Chem. Phys., 13, 11019-11058, doi:10.5194/acp-13-11019-2013, 2013.

# Acknowledgements

This work has been funded by the EU FP7 MarcoPolo project, www.marcopolo.eu, 2014-2017. Results presented in this work have been produced using the European Grid Infrastructure (EGI) through the National Grid Infrastructures NGI_GRNET (HellasGrid) as part of the SEE Virtual Organization. The authors would like to acknowledge the support provided by the Scientific Computing Office, IT A.U.Th., throughout the progress of this research work. We wholeheartedly thank Ass. Prof. Eleni Katragkou for her assistance with the ERA Interim datasets.

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
