# Peer review of "Updated SO2 emission estimates over China using OMI/Aura observations"

_Atmospheric Measurement Techniques, 2017_

## Referee Comment (RC1) · Anonymous Referee #2 · 20 Oct 2017

**1 Summary**

The manuscript presents an updated SO2 emission inventory for China using the MEIC v.12 inventory as an apriori and generating an aposteriori inventory using OMI/Aura SO2 observations and SO2 profiles from the CHIMERE CTM. The updated inventory shows new source areas in SW and SE China, which do not appear in the original MEIC inventory. The results show satisfying agreement with other emission inventories.

[Figure]

**2 General comments**

The paper is well written and all sources well referenced. The method used and the data sources are well described, however I have one specific question that I would like the authors to clarify:

In order to calculate the aposteriori emissions using the inversion methodology presented in section 3.1 the apriori emission field is multiplied by the satellite-derived $SO_2$ field divided by the model $SO_2$ field. In order to calculate the satellite-derived field from the OMI satellite observations, AMFs are calculated using an anthropogenic $SO_2$ profile from the IMAGES CTM. Why didn't the authors use the same $SO_2$ profile for the calculation of the satellite field (i.e. in the AMF calculation) AND the model $SO_2$ field? In this way one would exclude differences between the IMAGES and CHIMERE CTM when calculating the updated emission inventory.

**3 Specific comments**

Unfortunately all multiplot maps shown in the paper are far too small. This is especially the case for Fig 1,5 and 7. In order to increase the image size I would suggest to remove the lat/lon axis labels between the single maps since all show the same area. Furthermore for Fig 1, I would suggest to use a different color bar, using white as the color for zero emissions.

Abstract

In the abstract it is written that 'novel inversion techniques' are used, however a broadly used technique is used (according to the papers cited in Section 3.1) and there is no 'novel technique' presented in this manuscript. This is misleading and I would suggest

replacing 'novel' with 'state-of-the-art' or 'broadly used'.

Introduction

- Wording: Sulphur dioxide / Sulfur dioxide – I have found both in the paper. Please use only one notation and check the paper again

- Page 2, line 17: Please name sources for hydrogen sulfide

- Page 2, line 23: What are 'scheduled biomass burning events'? Please clarify

Section 2.2

- Page 5, line 11: Are daily/monthly/fixed SO2 profiles from the IMAGES CTM used? Please clarify

- Page 5, line 20: SO2 algorithm flagging: What exactly is flagged? Perhaps add a short list or example.

- Page 6, line 4/5: NS,0 is not used in any equation What is meant by SCD-SCD correction? Typo: AMD precision. I guess this should be AMF precision

Section 2.3

- Page 7, line 17/ Page 9, line 29/ Fig4: There is general confusion when using the terms layer or level throughout this section. What I understood is that the model provides SO2 vmr in ppm on nine (or eight???) levels from which SO2 partial columns in eight (or seven??) layers can be calculated. Hence Fig 4 is not correct – you can't show the SO2 profiles in ppb and DU on the same grid – for

the SO2 profile in DU the layer midpoints should be used and not the levels from the vmr

The text should be corrected accordingly:

- P.7, l 16/17: ...on nine vertical layers levels in ppb, i.e. seven vertical layers
- P.9, l 29 Fig. 4 – eight or nine levels for vmr? Please clarify!

Section 4.1

- Page 13. Line 24-26. This is not clear for me. Why did only a part of the 8414 grid cells actually provide information?

- Figure 6. One could also add the MEIC emissions for the years 2008,2010 and 2012 to the plots to get a better overview of the agreement in different years.

- Page 16, Line 16: It is unclear from the text that the increase for 2010 is wrt to the MEIC apriori inventory. Please clarify in the text

---

## Referee Comment (RC2) · Anonymous Referee #1 · 24 Oct 2017

This paper updates existing SO2 emission inventories over China by using OMI observations and CTM. New source areas missing from the bottom-up inventories are identified and SO2 emission trends are interpreted. However, it is not very easy for readers to follow the contents, in particular the methodology part. I strongly suggest the authors spend some time on improving this part.

General comments:

1. The introduction section needs to be improved. I suggest focusing on literatures related to authors' own work, instead of a very general introduction. The relationship between the previous studies and this work needs to be clarified. More recent work, e.g., Krotkov et al., 2016, van der A et al., 2017, needs to be included.

2. The method developed by Martin et al., 2003 works very well for NOx, because NOx lifetime is relatively short and it does not bring significant uncertainties by ignoring transport between grid cells. However, this is not the case for SO2. A further analysis is necessary to convince the method is still solid for SO2.

3. In section 4.2, the authors tabulate the significant differences between inventories, but without any explanations for the reasons. I suggest a similar analysis as conducted in your recent work (Ding et al., 2017) to explore the possible reasons.

Specific comments:

1. Page 2, line 14, the meaning of "usable manner" is confusing. Please consider rephrasing it.

2. Page 2, line 38, please consider rephrasing "emission fields".

3. Page 3, line 16, please state the reason for the given error of 50%.

4. Page 3, line 27, please clarify the reason why the emissions in "great Beijing areas" is best represented.

5. Figure 1. It is not easy to distinguish the differences between graphs using the current legend.

6. Page 8, line 3, please clarify the sources of the uncertainty of the CHIMERE SO2 columns.

7. Page 9, line 21, it is not accurate to say "the OMI observations are point daily measurements". The OMI observation cannot be treated as a "point".

8. Page 9, line 29. How many levels of CHMIERE output are used in this study? It says 8 here, but 7 before.

9. Page 9, line 30. What is the "OMI 58 AK levels"?

10. Page 17, line 19. What is the definition of "SO2 emission fields"?

---

## Author Comment (AC1) · 29 Dec 2017

We warmly thank for referee for her/his positive take on our work and helpful comments.

**General Comments**

The paper is well written and all sources well referenced. The method used and the data sources are well described, however I have one specific question that I would like the authors to clarify:

In order to calculate the aposteriori emissions using the inversion methodology presented in section 3.1 the apriori emission field is multiplied by the satellite-derived SO2 field divided by the model SO2 field. In order to calculate the satellite-derived field from the OMI satellite observations, AMFs are calculated using an anthropogenic SO2 profile from the IMAGES CTM. Why didn't the authors use the same SO2 profile for the calculation of the satellite field (i.e. in the AMF calculation) AND the model SO2 field? In this way one would exclude differences between the IMAGES and CHIMERE CTM when calculating the updated emission inventory.

The reviewer is raising a very interesting suggestion which might have been possible if the satellite field calculations and the CHIMERE CTM run where performed within the same operational chains. However, the former are produced in an operational manner by BIRA whereas the latter by KNMI. The suggestion of the reviewer would hence require the reprocessing of the satellite data, which is beyond the scope of this paper.

**Specific comments**

Unfortunately all multiplot maps shown in the paper are far too small. This is especially the case for Fig 1,5 and 7. In order to increase the image size I would suggest to remove the lat/lon axis labels between the single maps since all show the same area. Furthermore for Fig 1, I would suggest to use a different color bar, using white as the color for zero emissions.

Thank you for this comment, indeed you are right. Figures 1, 5, & 7 have been updated accordingly.

**Abstract**

In the abstract it is written that 'novel inversion techniques' are used, however a broadly used technique is used (according to the papers cited in Section 3.1) and there is no 'novel technique' presented in this manuscript. This is misleading and I would suggest replacing 'novel' with 'state-of-the-art' or 'broadly used'.

Line re-phrased.

**Introduction**

• Wording: Sulphur dioxide / Sulfur dioxide – I have found both in the paper. Please use only one notation and check the paper again

Sulphur dioxide was kept as notation.

• Page 2, line 17: Please name sources for hydrogen sulfide

Line added in the relevant section.

• Page 2, line 23: What are 'scheduled biomass burning events'? Please clarify

Basically, the burning of croplands in order to re-plant for the new season, i.e. the agriculture sector. Line added in the relevant section.

**Section 2.2**

• Page 5, line 11:  Are daily/monthly/fixed SO2 profiles from the IMAGES CTM used? Please clarify

Daily profiles were used, at the overpass time of OMI. Line added in the relevant section.

• Page 5, line 20: SO2 algorithm flagging: What exactly is flagged? Perhaps add a short list or example.

Wording altered.

• Page 6, line 4/5: NS,0 is not used in any equation What is meant by SCD-SCD correction? Typo: AMD precision. I guess this should be AMF precision

Thank you for being so attentive. The $N_{S,0}$ does not appear in these equations, indeed. The *SCD-SCD correction* is the Slant Density minus the Slant Density correction, and the AMD precision is indeed a typo.

**Section 2.3**

• Page 7, line 17/ Page 9, line 29/ Fig4: There is general confusion when using the terms layer or level throughout this section.  What I understood is that the model provides SO2 vmr in ppm on nine (or eight???)  levels from which SO2 partial columns in eight (or seven??) layers can be calculated. Hence Fig 4 is not correct – you can't show the SO2 profiles in ppb and DU on the same grid – for
the SO2 profile in DU the layer midpoints should be used and not the levels from the vmr. The text should be corrected accordingly:

– P.7, l 16/17: . . .on nine vertical layers levels in ppb, i.e. seven vertical layers
– P.9, l 29 Fig. 4 – eight or nine levels for vmr? Please clarify! Section 4.1

Thank you for this comment, indeed, we confused the terms *layer* and *level* in the text, it should be clear now. You are also correct on the depiction comment on ppb and DU, it was inadvertently plotted on the "wrong" altitude grid. The calculations were performed appropriately.

• Page 13. Line 24-26. This is not clear for me. Why did only a part of the 8414 grid cells actually provide information?

The domain studied is between 102° to 132°E and 15° to 55°N, on a 0.25x0.25° spacing, however the MEIC emission inventory covers only part of that domain, mainland China. As a result, only 8414 grid cells out of the possible 19200 can be analyzed.

• Figure 6. One could also add the MEIC emissions for the years 2008,2010 and 2012 to the plots to get a better overview of the agreement in different years.

This is a very good point. We are currently working towards a companion paper which will present the comparisons between the different emission inventories for SO2 over the region, as per Ding et al., 2007. First results were presented to the scientific community during the 18th GEIA conference in Hamburg in September 2017 [presentations online here: http://www.geiacenter.org/community/geia-conferences/2017-conference]. We hence feel that adding this material to Figure 6 of this paper would make it difficult to interpret, without all the supporting material already in the companion paper.

• Page 16, Line 16: It is unclear from the text that the increase for 2010 is wrt to the MEIC apriori inventory. Please clarify in the text

Wording altered.

---

## Author Comment (AC2) · 29 Dec 2017

First of all, we would like to warmly thank the reviewer for her/his time in improving our work through helpful and suggestive comments.

This paper updates existing SO2 emission inventories over China by using OMI observations and CTM. New source areas missing from the bottom-up inventories are identified and SO2 emission trends are interpreted. However, it is not very easy for readers to follow the contents, in particular the methodology part. I strongly suggest the authors spend some time on improving this part.

General comments:

1. The introduction section needs to be improved. I suggest focusing on literatures related to authors' own work, instead of a very general introduction. The relationship between the previous studies and this work needs to be clarified. More recent work, e.g., Krotkov et al., 2016, van der A et al., 2017, needs to be included.

Introduction expanded as requested.

2. The method developed by Martin et al., 2003 works very well for NOx, because NOx lifetime is relatively short and it does not bring significant uncertainties by ignoring transport between grid cells. However, this is not the case for SO2. A further analysis is necessary to convince the method is still solid for SO2.

The issue is known to the authors and we have long discussed it also with esteemed colleagues in the field. In contrast to the equivalent recent NOx emission estimates by the Martin technique [see for e.g. Zyrichidou et al., 2015[1]], we are working on a coarser 0.25x0.25 degree grid. However, since both the apriori emissions as well as the modelling inputs are on a monthly scale, we were unable to configure a way to quantify any smearing effect due to transport [daily effect]. Hains et al., 2008[2], provide a global scale estimate for the $SO_2$ lifetime to be 19 ± 7 h, while Fioletov et al, 2015[3], provide a range of 4h to 12h for the lifetime for $SO_2$. Other studies (Lee et al, 2011[4]) show even larger variability for the lifetime of $SO_2$, between 16 and 40h. Considering this large range of estimates for the lifetime of $SO_2$ we can only claim that our estimates should be valid for the lower lifetime estimates of $SO_2$ and of course this range of uncertainty in the $SO_2$ lifetime would be a main source of uncertainty in our aposteriori estimates. We have added an explanatory section at the end of section 2.2 on the matter.

3. In section 4.2, the authors tabulate the significant differences between inventories, but without any explanations for the reasons. I suggest a similar analysis as conducted in your recent work (Ding et al., 2017) to explore the possible reasons.

[1] Zyrichidou, I., M.E. Koukouli, D. Balis, K. Markakis, A. Poupkou, E. Katragkou, I. Kioutsioukis, D. Melas, K.F. Boersma, M. van Roozendael, Identification of surface NO emission sources on a regional scale using OMI NO, Atmospheric Environment, http://dx.doi.org/10.1016/j.atmosenv.2014.11.023.

[2] Hains, J. C., B. F. Tabumann, A. M. Thompson, J. W. Stehr, L. T. Marufu, B. G. Doddridge, and R. R. Dickerson (2008), Origins of chemical pollution derived from mid-Atlantic aircraft profiles using a clustering technique, Atmos. Environ., 42, 1727–1741, doi:10.1016/j.atmosenv.2007.11.052.

[3] Fioletov, V. E., C. A. McLinden, N. Krotkov, and C. Li (2015), Lifetimes and emissions of SO2 from point sources estimated from OMI. Geophys. Res. Lett., doi: 10.1002/2015GL063148, 2015.

[4] Lee, C., R. V. Martin, A. van Donkelaar, H. Lee, R. R. Dickerson, J. C. Hains, N. Krotkov, A. Richter, K. Vinnikov, and J. J. Schwab, SO2 emissions and lifetimes: Estimates from inverse modeling using in situ and global, space-based (SCIAMACHY and OMI) observations, J. Geophys. Res., doi:10.1029/2010JD014758, 2011.

This is indeed the next logical step in this work, one which we are already undertaking. First results were presented to the scientific community during the 18th GEIA conference in Hamburg in September 2017 [presentations online here: http://www.geiacenter.org/community/geia-conferences/2017-conference] and we are actively working on a comparison paper, following the logic of the work performed for NOx in Ding et al., 2017. However, we feel that adding this material to this paper would render it rather long and beyond the scope which is to introduce the new emission inventory.

Specific comments:

1. Page 2, line 14, the meaning of "usable manner" is confusing. Please consider rephrasing it.

You are correct, line simplified.

2. Page 2, line 38, please consider rephrasing "emission fields".

Line rephrased.

3. Page 3, line 16, please state the reason for the given error of 50%.

The MEIC inventory does not have an associated error estimate included and we were forced to assume one. In our new work, where the bottom-up and the top-down inventories are inter-compared in detail, we have performed sensitivity studies on the methodology by altering this value from a small estimate of 10% to a large estimate of 90% and will present the effect this has on the final updated emission inventory.

4. Page 3, line 27, please clarify the reason why the emissions in "great Beijing areas" is best represented.

Line added in the text.

5. Figure 1. It is not easy to distinguish the differences between graphs using the current legend.

We have altered the colour bars accordingly.

6. Page 8, line 3, please clarify the sources of the uncertainty of the CHIMERE $SO_2$ columns.

In the work of Beekmann and Derognat, 2003[5], and subsequently in Deguillaume, et al., 2007[6], a Bayesian Monte Carlo analysis was applied to the CHIMERE model over Paris in order to estimate the overall uncertainty with respect to the following CHIMERE model input parameters: anthropogenic and biogenic emissions, meteorological parameters such as wind speed and mixing layer height, actinic fluxes, quantum yields, and chemical rate coefficients. However, they only report assessments for tropospheric ozone, and then on secondary NOx and VOC formation, and not on $SO_2$. CHIMERE runs were also used to assess SCIAMACHY observations [Blond et al., 2007[7]]
* * *
[5] Beekmann, M., and C. Derognat (2003), Monte Carlo uncertainty analysis of a regional-scale transport chemistry model constrained by measurements from the Atmospheric Pollution Over the Paris Area (ESQUIF) campaign, J. Geophys. Res., doi:10.1029/2003JD003391.

[6] Deguillaume, L., M. Beekmann, and L. Menut (2007), Bayesian Monte Carlo analysis applied to regional-scale inverse emission modeling for reactive trace gases, J. Geophys. Res., 112, D02307, doi:10.1029/2006JD007518.

[7] Blond, N., K. F. Boersma, H. J. Eskes, R. J. van der A, M. Van Roozendael, I. De Smedt, G. Bergametti, and R. Vautard (2007), Intercomparison of SCIAMACHY nitrogen dioxide observations, in situ measurements and air quality modeling results over Western Europe, J. Geophys. Res., 112,

which includes error estimates but again for NO$_2$ only.

Within the framework of the EU FP7 MarcoPolo project, http://www.marcopolo-panda.eu/, an ensemble of modelled SO$_2$ estimates were inter-compared with in-situ observations and Figure 1 shows the relative percentage error of each model. During the OMI/Aura overpass time, CHIMERE has about 20-40% uncertainty SO$_2$ on surface concentration.

[Figure]

Figure 1. Inter-comparison of SO$_2$ estimates by different model runs [in different colours] to the CHIMERE estimate [red line]. From top to bottom: mean SO$_2$, STD SO$_2$, CORR SO$_2$ and RMS SO$_2$. **Unpublished results**.

7. Page 9, line 21, it is not accurate to say "the OMI observations are point daily measurements". The OMI observation cannot be treated as a "point".

You are of course correct, line re-phrased.

8. Page 9, line 29. How many levels of CHMIERE output are used in this study? It says 8 here, but 7 before.

Apologies, small typo error mixing up the words layers and levels. The entire text was checked and amended accordingly.

9. Page 9, line 30. What is the "OMI 58 AK levels"?

Phrase added.

10. Page 17, line 19. What is the definition of "SO2 emission fields"?

Wording rephrased throughout the text. We simply meant that we are producing an actual spatial domain, in lat/lon, of emissions and not total SO2 emitted masses over specific source locations.

D10311, doi:10.1029/2006JD007277.

---

## Author Comment (AC3) · 29 Dec 2017

The comment was uploaded in the form of a supplement:
https://www.atmos-meas-tech-discuss.net/amt-2017-256/amt-2017-256-AC3-supplement.pdf

———————————————————

---

## Author Response (AR1)

First of all, we would like to warmly thank the reviewer for her/his time in improving our work through helpful and suggestive comments.

This paper updates existing SO2 emission inventories over China by using OMI observations and CTM. New source areas missing from the bottom-up inventories are identified and SO2 emission trends are interpreted. However, it is not very easy for readers to follow the contents, in particular the methodology part. I strongly suggest the authors spend some time on improving this part.

**General comments:**

1. The introduction section needs to be improved. I suggest focusing on literatures related to authors' own work, instead of a very general introduction. The relationship between the previous studies and this work needs to be clarified. More recent work, e.g., Krotkov et al., 2016, van der A et al., 2017, needs to be included.

**Introduction expanded as requested.**

2. The method developed by Martin et al., 2003 works very well for NOx, because NOx lifetime is relatively short and it does not bring significant uncertainties by ignoring transport between grid cells. However, this is not the case for SO2. A further analysis is necessary to convince the method is still solid for SO2.

The issue is known to the authors and we have long discussed it also with esteemed colleagues in the field. In contrast to the equivalent recent NOx emission estimates by the Martin technique [see for e.g. Zyrichidou et al.,  $2015^1$ ], we are working on a coarser 0.25x0.25 degree grid. However, since both the apriori emissions as well as the modelling inputs are on a monthly scale, we were unable to configure a way to quantify any smearing effect due to transport [daily effect]. Hains et al.,  $2008^2$ , provide a global scale estimate for the SO2 lifetime to be  $19 \pm 7$  h, while Fioletov et al,  $2015^3$ , provide a range of 4h to 12h for the lifetime for SO2. Other studies (Lee et al,  $2011^4$ ) show even larger variability for the lifetime of SO2, between 16 and 40h. Considering this large range of estimates for the lifetime estimates of SO2 and of course this range of uncertainty in the SO2 lifetime would be a main source of uncertainty in our aposteriori estimates. We have added an explanatory section at the end of section 2.2 on the matter.

3. In section 4.2, the authors tabulate the significant differences between inventories, but without any explanations for the reasons. I suggest a similar analysis as conducted in your recent work (Ding et al., 2017) to explore the possible reasons.

<sup>1 Zyrichidou, I., M.E. Koukouli, D. Balis, K. Markakis, A. Poupkou, E. Katragkou, I. Kioutsioukis, D. Melas, K.F. Boersma, M. van Roozendael, Identification of surface NO emission sources on a regional scale using OMI NO, Atmospheric Environment, http://dx.doi.org/10.1016/j.atmosenv.2014.11.023.

<sup>2 Hains, J. C., B. F. Tabumann, A. M. Thompson, J. W. Stehr, L. T. Marufu, B. G. Doddridge, *and* R. R. Dickerson (2008), Origins of chemical pollution derived from mid-Atlantic aircraft profiles using a clustering technique, Atmos. Environ., 42, 1727–1741, *doi*:10.1016/j.atmosenv.2007.11.052. 3 Fioletov, V. E., C. A. McLinden, N. Krotkov, and C. Li (2015), Lifetimes and emissions of SO2 from point sources estimated from OMI. Geophys. Res. Lett., *doi*: 10.1002/2015GL063148, 2015.

<sup>4 Lee, C., R. V. Martin, A. van Donkelaar, H. Lee, R. R. Dickerson, J. C. Hains, N. Krotkov, A. Richter, K. Vinnikov, and J. J. Schwab, SO2 emissions and lifetimes: Estimates from inverse modeling using in situ and global, space-based (SCIAMACHY and OMI) observations, J. Geophys. Res., doi:10.1029/2010JD014758, 2011.

This is indeed the next logical step in this work, one which we are already undertaking. First results were presented to the scientific community during the 18th GEIA conference in Hamburg in September 2017 [presentations online here: http://www.geiacenter.org/community/geiaconferences/2017-conference] and we are actively working on a comparison paper, following the logic of the work performed for NOx in Ding et al., 2017. However, we feel that adding this material to this paper would render it rather long and beyond the scope which is to introduce the new emission inventory.

Specific comments:

1. Page 2, line 14, the meaning of "usable manner" is confusing. Please consider rephrasing it.

You are correct, line simplified.

2. Page 2, line 38, please consider rephrasing "emission fields".

**Line rephrased.**

3. Page 3, line 16, please state the reason for the given error of 50%.

The MEIC inventory does not have an associated error estimate included and we were forced to assume one. In our new work, where the bottom-up and the top-down inventories are intercompared in detail, we have performed sensitivity studies on the methodology by altering this value from a small estimate of 10% to a large estimate of 90% and will present the effect this has on the final updated emission inventory.

4. Page 3, line 27, please clarify the reason why the emissions in "great Beijing areas" is best represented.

**Line added in the text.**

5. Figure 1. It is not easy to distinguish the differences between graphs using the current legend.

We have altered the colour bars accordingly.

6. Page 8, line 3, please clarify the sources of the uncertainty of the CHIMERE SO2 columns.

In the work of Beekmann and Derognat, 20035, and subsequently in Deguillaume, et al., 20076, a Bayesian Monte Carlo analysis was applied to the CHIMERE model over Paris in order to estimate the overall uncertainty with respect to the following CHIMERE model input parameters: anthropogenic and biogenic emissions, meteorological parameters such as wind speed and mixing layer height, actinic fluxes, quantum yields, and chemical rate coefficients. However, they only report assessments for tropospheric ozone, and then on secondary NOx and VOC formation, and not on SO2. CHIMERE runs were also used to assess SCIAMACHY observations [Blond et al., 20077]

<sup>5 Beekmann, M., and C. Derognat (2003), Monte Carlo uncertainty analysis of a regional-scale transport chemistry model constrained by measurements from the Atmospheric Pollution Over the Paris Area (ESQUIF) campaign, J. Geophys. Res., doi:10.1029/2003JD003391.

<sup>6 Deguillaume, L., M. Beekmann, and L. Menut (2007), Bayesian Monte Carlo analysis applied to regional-scale inverse emission modeling for reactive trace gases, J. Geophys. Res., 112, D02307, doi:10.1029/2006JD007518.

<sup>7 Blond, N., K. F. Boersma, H. J. Eskes, R. J. van der A, M. Van Roozendael, I. De Smedt, G. Bergametti, and R. Vautard (2007), Intercomparison of SCIAMACHY nitrogen dioxide observations, in situ measurements and air quality modeling results over Western Europe, J. Geophys. Res., 112,

which includes error estimates but again for NO2 only.

Within the framework of the EU FP7 MarcoPolo project, http://www.marcopolo-panda.eu/, an ensemble of modelled SO2 estimates were inter-compared with in-situ observations and Figure 1 shows the relative percentage error of each model. During the OMI/Aura overpass time, CHIMERE has about 20-40% uncertainty SO2 on surface concentration.

---

## Editor Decision (ED1)

Dear authors

Please correct the following :

- P1, L22 where → were
- P6, L15 van der A et al., (2016) → van der A et al. (2017)
- P8, L7 have been already been → have already been
- P13, L8 formulism → formalism (??? Not sure please check)
- P27, L27-28 Disc. Doi.10.5194/acp-2016-445 → doi:10.5194/acp-17-1775-2017

Ilse Aben, co-editor